cognition

numerical equivalence, mathematical cognition, animal cognition, cognitive development, non-human primate

**Author for correspondence:**
Sarah E. Koopman
e-mail: skoopman@ur.rochester.edu

# One-to-one correspondence without language

Sarah E. Koopman[1], Alyssa M. Arre[2],
Steven T. Piantadosi[1,3] and Jessica F. Cantlon[1,4]

[1]Brain and Cognitive Sciences, University of Rochester, 500 Wilson Boulevard, Rochester, NY, USA
[2]Psychology, Yale University, New Haven, CT, USA
[3]Psychology, University of California, Berkeley, CA, USA
[4]Psychology, Carnegie Mellon University, Pittsburgh, PA, USA

SEK, 0000-0001-8116-9913; AMA, 0000-0002-1974-5067

A logical rule important in counting and representing exact number is one-to-one correspondence, the understanding that two sets are equal if each item in one set corresponds to exactly one item in the second set. The role of this rule in children's development of counting remains unclear, possibly due to individual differences in the development of language. We report that non-human primates, which do not have language, have at least a partial understanding of this principle. Baboons were given a quantity discrimination task where two caches were baited with different quantities of food. When the quantities were baited in a manner that highlighted the one-to-one relation between those quantities, baboons performed significantly better than when one-to-one correspondence cues were not provided. The implication is that one-to-one correspondence, which requires intuitions about equality and is a possible building block of counting, has a pre-linguistic origin.

## 1. Introduction

The formal mathematics critical in STEM fields relies on the ability to represent exact quantities. However, learning the concept of exact number takes considerable time and effort. Humans and other animals possess two core cognitive systems that can represent quantities—the approximate number system (ANS) and the parallel individuation system [1]—neither of which can represent large quantities exactly. The ANS represents quantities roughly [2,3] and the parallel individuation system can represent exact quantities, but only up to three or four [4–6]. Although children as young as 2 years old can learn the count list of numbers, it is not until they are about 3.5 years old that they truly understand that each number word refers to a particular exact quantity [7] and perhaps even longer before they have a complete concept of exact number [8].

How children acquire exact number from the cognitive systems available to them (ANS, parallel individuation and

count list) remains unclear, but one key element seems to be the logical ability of one-to-one correspondence. This ability entails comparing two sets by pairing each item in one set with exactly one item in the second set. Gelman & Gallistel [9] argue that one-to-one correspondence is necessary to understand the logic of counting, as it enables the counter to pair one number word with each item in the set being enumerated. They contend that one-to-one correspondence is inherent in the accumulator model hypothesized to underlie the ANS [10]. One-to-one correspondence is also thought to be inherent in parallel individuation, whereby mentally stored items in one set can be compared one-to-one with visible objects to detect numerical matches or mismatches [11–13]. Exact numerical equality, a feature of the integer system, is thought to rely on an understanding of one-to-one correspondence: if the items of two sets can be placed in one-to-one correspondence with each other, then those sets are quantitatively equal [14,15]. Finally, all models of counting development assume that children use one-to-one correspondence to count [16–18], implying that it is both available to young children and a necessary precursor to counting.

Although one-to-one correspondence can be used to pair items with counting words and establish numerical equivalence, it is neither symbolic nor inherently quantitative, aside from requiring a representation of exactly one [19], suggesting that an understanding of one-to-one correspondence is possible prior to counting. However, research investigating one-to-one correspondence ability in non-counting populations has yielded mixed results. At least a partial understanding of one-to-one correspondence has been shown by infants [20,21], pre-counters [14,22] and populations without exact number words [19,23,24]. Yet in many of these cases, individuals succeed in using one-to-one correspondence in some contexts but not others. For example, 3-year-olds succeeded in reconstructing sets of five or six objects using one-to-one correspondence, unless an object in the set was visibly added or removed [14]. Some non-counting populations appear to match only small sets exactly [25–27]. In other studies, non-counters do not show any understanding of one-to-one correspondence [28–30]. For example, young children were shown two identical sets of images, with the images arranged in lines to highlight the one-to-one correspondence of the images in the two sets. When told how many images one set contained, the children could not correctly label the second set with the same number word [30]. Interestingly, the tasks used in these studies where children failed to use one-to-one correspondence all involved the use of number words, making it unclear whether non-counters' failure was due to lack of one-to-one correspondence ability or incomplete understanding of number words.

The role of one-to-one correspondence in the development of exact number thus remains unclear. Potential confounding factors include individual differences in development, particularly in the development of language. Moreover, the origin of the logical ability of one-to-one correspondence is unknown; in fact, it is unclear whether this ability is unique to humans or is shared with other animals. One study examining quantity discrimination ability in canids used a procedure that provided cues to one-to-one correspondence between the quantities, but results indicated that these cues did not enhance the canids' discrimination performance [31]. No other studies have investigated one-to-one correspondence ability in animals.

A comparative approach with non-human primates could help tease apart the role of language and other cognitive abilities in the development of exact number. Non-human primates do not have language, and only learn an ordered list of symbols (a non-linguistic 'count list') with training. We can, therefore, investigate whether one-to-one correspondence can be understood without a count list, while controlling for the effects of language. This will provide insight into whether some of the logic that is essential to human counting has a primitive origin, existing prior to or in the absence of verbal counting. Non-human primates have previously been shown to use arithmetic reasoning like addition, subtraction and proportion to compare quantities [32–38], as well as logic like transitive inference and mutual exclusivity [39–41], so it is reasonable to expect that they might be able to use a one-to-one correspondence relation to recognize equality between sets.

Here, we use a quantity discrimination task to examine whether olive baboons (*Papio anubis*) can use one-to-one correspondence to make more accurate numerical discriminations than predicted by the precision of their ANS. In traditional quantity discrimination tasks, the quantities to be discriminated are presented in a sequential manner (all the items in one set are presented one-by-one before the items in the second set are presented one-by-one). This presentation method requires that the subject continuously track the quantity of items in the sets. As the ratio between the quantities being compared increases, accuracy decreases, a signature of the ANS due to its imprecise nature.

We compare performance on these trials with performance on trials where quantities are baited such that one-to-one correspondence can be easily used to compare the quantities (items are presented one-by-one in both sets simultaneously, then an additional item is added to or removed from one set). One-to-one

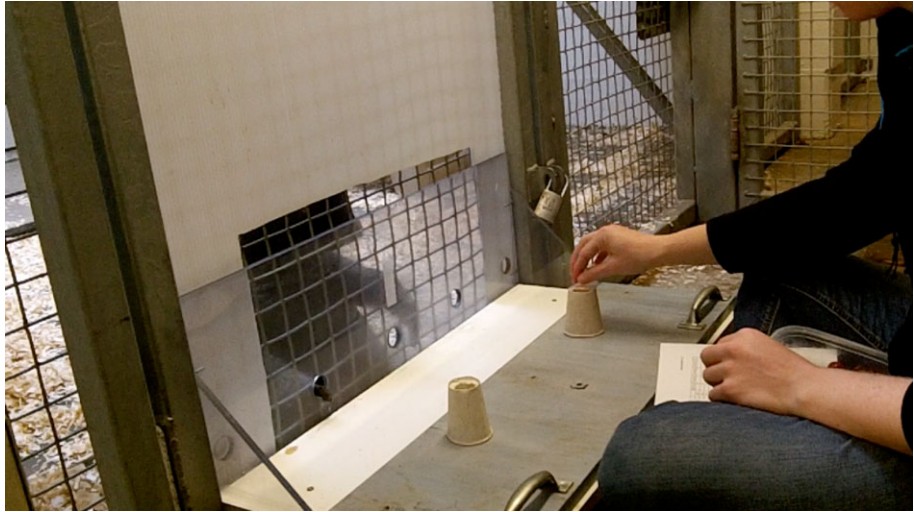

**Figure 1.** Experimental apparatus.

correspondence is a logical principle not reliant on magnitude, so its use should not have the limits of the ANS. Subjects can simply represent equality in the one-to-one relation and then recognize that one set receives (or loses) an additional item. We leverage this important difference between the ANS and one-to-one correspondence by testing quantity pairs with ratios that are difficult to discriminate using the ANS. Accurate discrimination of quantities in the one-to-one condition that cannot be discriminated in the sequential condition would suggest an understanding of one-to-one correspondence.

# 2. Methods

## 2.1. Subjects

Five olive baboons (*Papio anubis*) housed and tested at the Seneca Park Zoo in Rochester, NY, participated in these experiments. Two baboons participated in all three experiments, one baboon only participated in Experiments 1 and 2 and two baboons only participated in Experiment 3. Primate chow and fresh fruits and vegetables are provided every morning, and water is available ad libitum. Research with these subjects was approved by the Seneca Park Zoo Conservation & Research Committee.

Subjects had prior training in quantity discrimination, but not with the one-to-one baiting procedure [42]. Subjects were reinforced with food on every trial and thus were not conditioned to discriminate quantity.

The sample size was determined by the number of subjects who could reach criterion in the training phase (out of 12 total baboons in the troop), indicating that they understood the task procedure. The small sample size is not unusual for primate studies and is, in fact, sufficient for our goal of investigating whether baboons have the capacity for understanding one-to-one correspondence. Establishing a capacity for an ability, unlike extrapolating an ability to a population, requires just one example.

## 2.2. Apparatus

The testing apparatus consisted of a low rectangular table shielded on one long side by plexiglass to prevent baboons from directly accessing items placed on the table (figure 1). There were five equally spaced ports in the plexiglass, which subjects could use to indicate their choices. During the experiment, the long side of plexiglass was pushed flush with the mesh of the enclosure, giving the subject access to the ports in the plexiglass. The subject sat behind the plexiglass (and mesh) and the experimenter sat opposite the subject. In order to partly obscure the experimenter to avoid experimenter cuing, an opaque corrugated plastic sheet in the shape of an upside down 'U' was placed between the plexiglass and the mesh. The opening in the plastic allowed access to the centre three holes in the plexiglass and extended above the top of the plexiglass by about one foot.

All experimental manipulations were conducted on a sliding panel that sat atop the table and that measured about half the depth of the table. The experimenter conducted all manipulations with the

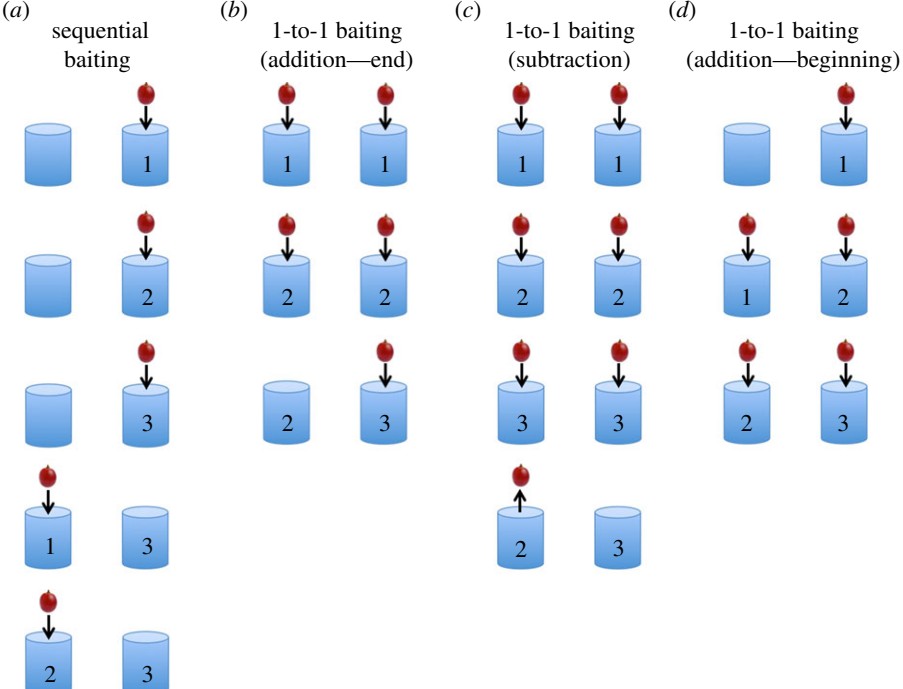

**Figure 2.** Baiting procedure in the (*a*) sequential condition, (*b*) one-to-one addition condition (extra added at the end), (*c*) one-to-one subtraction condition and (*d*) one-to-one addition condition (extra added at the beginning).

panel placed close to the experimenter, then pushed the panel forward towards the subject to indicate that the manipulations were over and that the subject should make a choice. The contents of the panel were two identical opaque cylinders made of either cardstock or plastic pipe, placed upright in front of the two outside unobscured ports in the plexiglass shield. The cylinders were open on both ends, allowing the experimenter to drop items into a cylinder and also lift a cylinder up, leaving the contents of the cylinder on the panel. We used the same food within each trial, and foods were either grapes, nuts or cereal. All animals liked to eat all of these foods. There were no trials where animals did not eat their rewards. A more detailed description of the apparatus is found in [42].

## 2.3. General procedure

Each session was conducted by two experimenters. One experimenter worked the apparatus, while a second experimenter recorded the choices made by the subject, monitored the first experimenter for trial accuracy and operated a video camera, which was used to record each session. Sessions were conducted when a subject could be temporarily isolated from the troop in an enclosure, which was between one and three times a week per baboon. Each session lasted approximately 30 min.

On sequential baiting trials, one cylinder was baited one item at a time, then the other cylinder was baited one item at a time (figure 2*a*). Each food item was shown to the subject for about 2 s before being placed into a cylinder, with about 2 s delay between items. To ensure that subjects were not basing their choices on the spatial location of the sets, the larger and smaller numerical values were equally likely to appear in any one of the two cylinders across the session. Similarly, to ensure that subjects were not basing their choices on the order of the sets, the larger and smaller numerical values were equally likely to be baited first.

On one-to-one addition trials, cylinders were baited simultaneously, one item at a time, with the baiting of the larger quantity continuing alone once the smaller quantity had been baited (figure 2*b*). To ensure that subjects were not basing their choices merely on which cup the experimenter last interacted with, one of the cylinders (randomly chosen) was lightly tapped on the top corner. On one-to-one subtraction trials, cylinders were baited simultaneously, one item at a time, up to the larger quantity (figure 2*c*). One or two items were then removed (one item at a time) from one cylinder and one of the cylinders was tapped. As with the sequential trials, the larger and smaller numerical values were equally likely to appear in either of the two cylinders.

To further examine whether subjects were using a lower-level strategy of choosing the cup that was baited last, one-to-one addition trials were conducted where the additional food piece was placed into one of the cups at the beginning of the trial rather than the end (figure 2d).

After the cylinders had been baited with food items, the panel was pushed forward and the subject was allowed to make a choice from between the two cylinders by sticking their finger through the corresponding port. Choices made before the panel was pushed forward were ignored. After the choice was made, the experimenter removed the cylinder from over the desired food and the food reward was fed, one item at a time, to the subject through the same port. The experimenter then removed the other cylinder from the panel, revealing its contents, and removed this set of reward. Once all food items were removed from the board, the experimenter pulled the panel back to her side of the apparatus and reset the board. The next trial was initiated. This procedure was used throughout the experiment.

## 2.4. Experiment 1

### 2.4.1. One versus two sequential training

In order to reacquaint the baboons with the quantity discrimination task and determine which baboons would participate for an entire session, a one versus two sequential task was conducted. All the comparisons made in this task were between one food item and two food items. Trials were counterbalanced for cylinder with the larger quantity, and the order of test trials was randomized within and between subjects. Only once a subject had completed two sessions in a row with at least 70% accuracy did s/he move to the next stage of training.

### 2.4.2. One versus two one-to-one addition training

This stage of training was identical to the one versus two sequential stage except that the cylinders were baited simultaneously. Trials were counterbalanced for cylinder with the larger quantity and cylinder tapped. The order of test trials was randomized within and between subjects. Only once a subject had completed two sessions in a row with at least 70% accuracy did s/he move to the testing phase.

### 2.4.3. One-to-one addition testing

Baboons were first tested on the quantity pairs 3 versus 4 and 4 versus 5, baited sequentially and one-to-one. Trials were counterbalanced for cylinder with the larger quantity, and the order of test trials was randomized within and between subjects. After completing testing on these quantity pairs (30–48 trials per pair one-to-one; 29–49 trials per pair sequential), baboons were then tested on the quantity pair 5 versus 6, baited sequentially and one-to-one (35–36 trials per condition). Trials were randomized and counterbalanced as above. Two baboons were tested on sequential and addition trials in separate sessions and one baboon was tested with sequential and addition trials interspersed.

## 2.5. Experiment 2

### 2.5.1. One-to-one subtraction testing

Following the conclusion of Experiment 1, the same three baboons were tested on subtraction. Baboons were tested on the quantity pairs 2 versus 4, 3 versus 4, 3 versus 5, 4 versus 5, 4 versus 6 and 5 versus 6 (subtraction) and 3 versus 4, 4 versus 5 and 5 versus 6 (addition). Addition and subtraction trials were interspersed to discourage baboons from basing their choices merely on which cup the experimenter last interacted with. Subjects completed 10–19 trials of each quantity pair in each condition.

## 2.6. Experiment 3

### 2.6.1. One versus two sequential training

Over a year after the conclusion of Experiment 2, in order to reacquaint the baboons with the quantity discrimination task and determine which baboons would participate for an entire session, a one versus two sequential task was conducted. All the comparisons made in this task were between one food item and two food items. Trials were counterbalanced for cylinder with the larger quantity, and

the order of test trials was randomized within and between subjects. Only once a subject had completed two sessions in a row with at least 70% accuracy did s/he move to the next stage of training.

### 2.6.2. One versus two addition training (baited last control)

This stage of training was identical to the one versus two sequential stage except that the cylinders were baited simultaneously. The additional food piece was added either at the beginning or at the end of the trial. These two types of trials were presented in a blocked fashion, with six trials of one type (e.g. extra added at beginning) followed by six trials of the other type (e.g. extra added at the end). Each session consisted of two blocks of each type for a total of 24 trials; blocks always alternated between the two types of trials, and the first block tested in a session was randomized between the two types of trials.

Trials were counterbalanced for cylinder with the larger quantity. The order of test trials was randomized within and between subjects. Only once a subject had completed two sessions in a row with at least 70% accuracy did s/he move to the testing phase. One baboon was mistakenly moved on to the testing phase after only one session with above-criterion accuracy.

### 2.6.3. One-to-one addition testing (baited last control)

To ensure the attention and motivation of the baboons, each test session began with a warm-up consisting of two blocks of one versus two addition trials (one block each of the extra food piece added at the beginning and at the end). If the baboon responded correctly on at least four of the six trials in both blocks, s/he received two blocks of test trials. Baboons were tested on the quantities 3 versus 4, 4 versus 5 and 5 versus 6. In one block, the extra food piece was added at the beginning of the trial, and in the other block, the extra food piece was added at the end of the trial. As in the one versus two addition training sessions, the four blocks in the test session alternated between these two types of trials. Trials were randomized and counterbalanced within this block structure. Subjects completed 19–21 trials of each quantity pair in each condition.

## 2.7. Data coding

The animals' choices were coded offline by one coder. Discrepancies between the coder and the experimenter were resolved by S.E.K. Two trials were excluded from analyses due to experimenter baiting error (6 versus 7), and three trials were excluded due to lack of video. A subset of the Experiment 1 and 2 sessions was also coded for the time taken to bait the cups in each trial (from when the first item was shown until the panel was pushed towards the subject; 26/50 sessions).

# 3. Results

## 3.1. Experiment 1: one-to-one addition

We tested baboons on quantity pairs that are difficult to discriminate using their ANS (as is required in the sequential baiting condition) to see if they could use one-to-one correspondence to discriminate those same quantities when baited in a manner that highlighted the one-to-one relation between those quantities (as can be done in the one-to-one addition condition). Figure 3 shows that baboons performed significantly above chance in the one-to-one addition condition (overall accuracy = 66.9%, $p < 0.001$; Pearl: 83 out of 131 trials, $p = 0.001$; Peperella: 89 out of 132 trials, $p < 0.001$; Sabina: 69 out of 97 trials, $p < 0.001$). Baboons also performed significantly above chance overall in the sequential condition (overall accuracy = 56.3%, $p = 0.01$; Pearl: 79 out of 132 trials, $p = 0.01$; Peperella: 69 out of 132 trials, $p = 0.33$; Sabina: 54 out of 95 trials, $p = 0.11$). Moreover, accuracy was significantly better on one-to-one trials than sequential trials, $\chi^2(1, N = 719) = 8.66$, $p = 0.003$, suggesting that highlighting the one-to-one relation between quantities facilitated discrimination. Logistic regressions revealed no significant effect of ratio on accuracy in either condition—addition: $b = -4.81$, OR = 0.008 [$8.69 \times 10^{-6}$, 6.84], $p = 0.16$ and sequential: $b = 0.46$, OR = 1.59 [0.003, 932.04], $p = 0.89$.

The effect of the final cup tap was examined in one-to-one addition trials using a logistic regression of accuracy on whether the correct cup was tapped in a particular trial. Tapping the correct cup did not significantly increase the log odds of choosing the correct cup, $b = 0.16$, OR = 1.17 [0.76, 1.83], $p = 0.47$, suggesting that the baboons did not base their choice on the last cup interacted with by the experimenter.

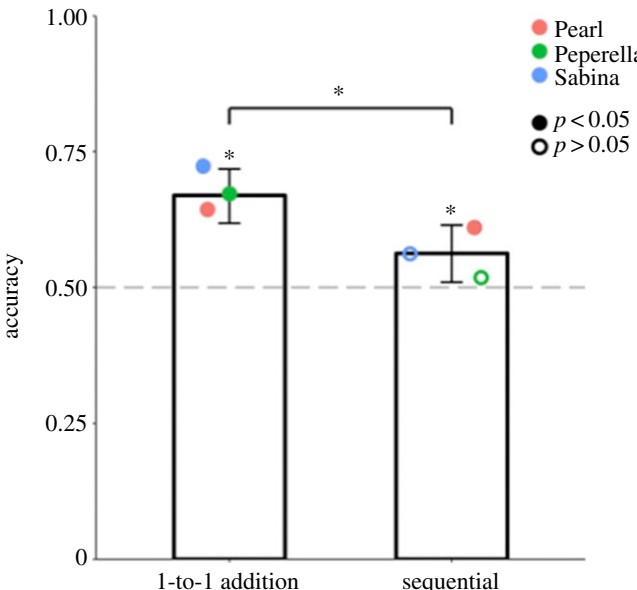

**Figure 3.** Accuracy in Experiment 1 for each condition. Bars indicate overall accuracy, while coloured circles correspond to individual baboons. Grey dashed line denotes the chance level. Error bars are 95% confidence intervals of two-tailed exact binomial tests. Filled circles denote accuracy significantly above chance; empty circles indicate non-significant accuracy. *$p < 0.05$.

To further investigate the possibility that the baboons merely chose the last cup baited by the experimenter, we used a logistic regression of the cup chosen in a trial on the cup that was baited last in that trial, the cup with more food pieces, whether the cup was baited sequentially or one-to-one (baiting condition), and the interaction between the cup baited last and the baiting condition. There was a significant main effect of the cup with more food pieces, $b = 0.82$, OR = 2.27 [1.39, 3.77], $p = 0.001$, but no main effects of the last cup baited, $b = 0.61$, OR = 1.85 [0.94, 3.60], $p = 0.07$, or condition, $b = -0.49$, OR = 0.61 [0.35, 1.06], $p = 0.08$. There was a significant interaction between condition and last cup baited, $b = 1.67$, OR = 5.33 [2.20, 13.54], $p < 0.001$; baboons were more likely to choose the cup that was baited last in the sequential condition (75%) than the one-to-one condition (67%). This analysis shows that baboons' choices were based largely on which cup contained more food pieces and that a strategy of choosing the cup baited last was actually less likely to be used when the cups were baited one-to-one than when they were baited sequentially.

### 3.2. Experiment 2: one-to-one subtraction

Figure 4 shows that, overall, baboons performed significantly above chance in both the one-to-one subtraction (accuracy = 67.9%, $p < 0.001$) and addition (accuracy = 64.7%, $p < 0.001$) conditions (Pearl: addition: 15 out of 30 trials, $p = 1$, subtraction: 46 out of 90 trials, $p = 0.46$; Peperella: addition: 25 out of 36 trials, $p = 0.01$, subtraction: 73 out of 107 trials, $p < 0.001$; Sabina: addition: 26 out of 36 trials, $p = 0.006$, subtraction: 88 out of 108 trials, $p < 0.001$; figure 4). There was no significant difference in accuracy between the one-to-one subtraction and addition conditions, $\chi^2(1, N = 407) = 0.35$, $p = 0.56$, suggesting that the baboons were not relying on a single, 'choose the last cup baited' strategy in 1-to-1 trials. A logistic regression revealed no significant effect of ratio on accuracy in either condition—addition: $b = -11.29$, OR = $1.25 \times 10^{-5}$ [$2.19 \times 10^{-11}$, 3.55], $p = 0.08$ and subtraction: $b = -0.43$, OR = 0.65 [0.08, 5.19], $p = 0.69$. There was also no significant effect of whether the correct cup was tapped on accuracy, $b = 0.13$, OR = 1.14 [0.73, 1.76], $p = 0.56$. Within the subtraction condition, the number of food pieces subtracted (1 or 2) did not affect accuracy, $b = 0.11$, OR = 1.12 [0.69, 1.81], $p = 0.65$.

Since the subtraction condition contained trials where the quantities differed by two, which were not present in the addition condition (where all quantities differed by one), the above analyses were also conducted excluding these potentially easier trials. As above, there was no significant difference in accuracy between the one-to-one subtraction and addition trials, $\chi^2(1, N = 255) = 0.10$, $p = 0.75$. The univariate logistic regressions also yielded the same results (table 1).

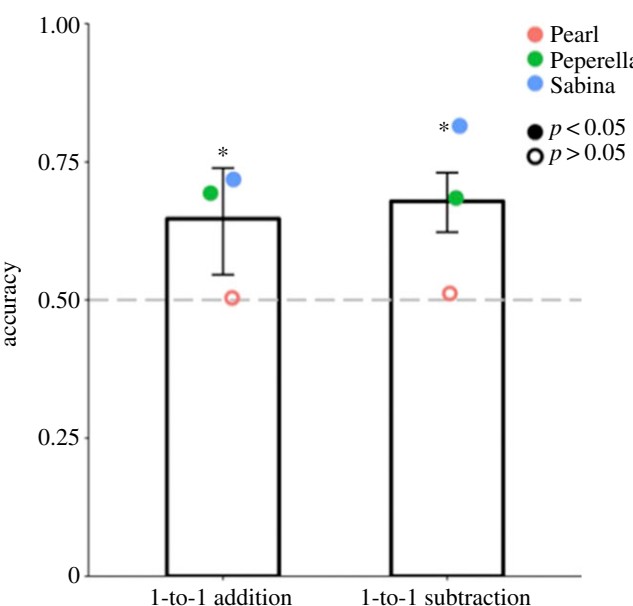

**Figure 4.** Accuracy in Experiment 2 for each condition. Bars indicate overall accuracy, while coloured circles correspond to individual baboons. Grey dashed line denotes the chance level. Error bars are 95% confidence intervals of two-tailed exact binomial tests. Filled circles denote accuracy significantly above chance; empty circles indicate non-significant accuracy. $*p < 0.05$.

**Table 1.** Results of univariate logistic regression analyses of factors related to performance in Experiment 2, excluding trials where two items were subtracted. CI, confidence interval.

| predictor | $b$ | s.e. $b$ | odds ratio | 95% CI |
|---|---|---|---|---|
| ratio (subtraction only) | −0.36 | 5.23 | 0.69 | $[2.20 \times 10^{-5}, 19\ 153.78]$ |
| tap correct | −0.04 | 0.28 | 0.97 | [0.55, 1.67] |

**Table 2.** Results of logistic regression analyses of the effect of trial condition on accuracy across Experiments 1 and 2. Subt, 1-to-1 subtraction condition; Add, 1-to-1 addition condition; Seq, sequential condition; CI, confidence interval.

| | predictor | $b$ | s.e. $b$ | odds ratio | 95% CI |
|---|---|---|---|---|---|
| all subtraction trials | Subt (versus Seq) | 0.50** | 0.16 | 1.64 | [1.20, 2.26] |
| | Add (versus Seq) | 0.45** | 0.15 | 1.57 | [1.16, 2.13] |
| | Subt (versus Add) | 0.04 | 0.17 | 1.04 | [0.75, 1.45] |
| subtract 1 trials only | Subt (versus Seq) | 0.44* | 0.20 | 1.55 | [1.05, 2.32] |
| | Add (versus Seq) | 0.45** | 0.15 | 1.57 | [1.16, 2.13] |
| | Subt (versus Add) | −0.01 | 0.20 | 0.99 | [0.66, 1.48] |

$*p < 0.05$; $**p < 0.01$.

## 3.3. Experiments 1 and 2 overall

Performance on the one-to-one subtraction trials in Experiment 2 was compared with performance on the sequential and one-to-one addition trials in Experiment 1. A logistic regression of accuracy on trial type showed significantly higher log odds of choosing the correct cup in either of the one-to-one conditions than the sequential condition; there was no significant difference between the one-to-one conditions (table 2). These results held when the subtraction trials with two food pieces were excluded from analyses, so that the analyses were conducted over the exact same pairs in all three conditions (table 2).

The effect of baiting time on accuracy was examined in a subset of trials (602/1126) using a multiple logistic regression of accuracy on trial condition (one-to-one or sequential) and baiting time. Only a main

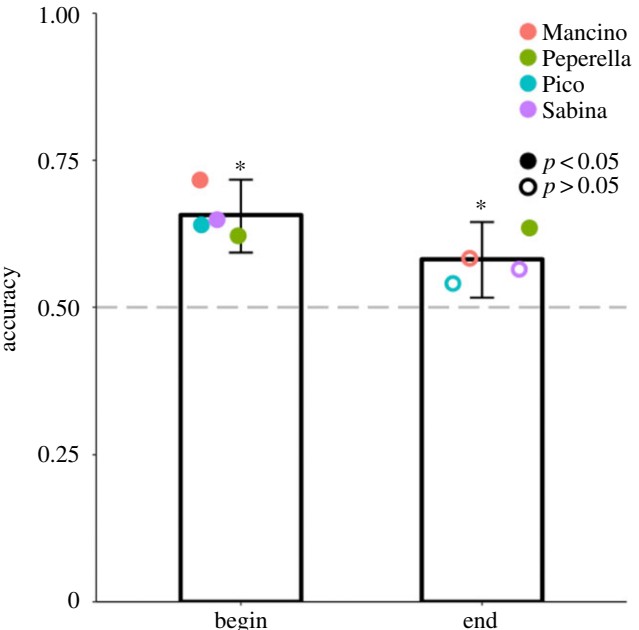

**Figure 5.** Accuracy in Experiment 3 for each condition. Bars indicate overall accuracy, while coloured circles correspond to individual baboons. Grey dashed line denotes the chance level. Error bars are 95% confidence intervals of two-tailed exact binomial tests. Filled circles denote accuracy significantly above chance; empty circles indicate non-significant accuracy. *$p < 0.05$.

effect of trial condition was significant, indicating that baboons had a significantly higher log odds of correctly responding to one-to-one trials than sequential trials when accounting for baiting time, $b = 0.58$, OR = 1.79 [1.25, 2.58], $p = 0.002$. There was no effect of baiting time, $b = -0.001$, OR = 1.00 [0.97, 1.02], $p = 0.94$, showing that baboons' higher accuracy on one-to-one trials was not due to trial length.

## 3.4. Experiment 3

Figure 5 shows that, overall, baboons performed significantly above chance when the extra food piece was added at the beginning (accuracy = 65.7%, $p < 0.001$) and the end (accuracy = 58.2%, $p = 0.007$) of a trial (Mancino: begin: 43 out of 60 trials, $p < 0.001$, end: 35 out of 60 trials, $p = 0.12$; Peperella: begin: 36 out of 58 trials, $p = 0.04$, end: 38 out of 60 trials, $p = 0.03$; Pico: begin: 39 out of 61 trials, $p = 0.02$, end: 32 out of 59 trials, $p = 0.30$; Sabina: begin: 39 out of 60 trials, $p = 0.01$, end: 34 out of 60 trials, $p = 0.18$; figure 4). There was no significant difference in accuracy when the extra food piece was added at the beginning or end of the trial, overall, $\chi^2(1, N = 476) = 2.87$, $p = 0.09$ and individually, Mancino: $\chi^2(1, N = 120) = 2.34$, $p = 0.13$; Peperella: $\chi^2(1, N = 118) = 0.02$, $p = 0.89$; Pico: $\chi^2(1, N = 120) = 1.17$, $p = 0.28$; Sabina: $\chi^2(1, N = 120) = 0.87$, $p = 0.35$.

## 3.5. Overall regression analysis

Figure 6 shows an overall analysis of all experiments. In this, we constructed a regression that predicted accuracy from condition coded across all experiments. The first bar (overall 1–1) shows a regression with accuracy collapsing across all one-to-one conditions in the three experiments, compared with the sequential condition. The three additional one-to-one bars ('1–1 add begin', '1–1 add end', '1–1 subtract') show a regression where the one-to-one conditions are separated and estimated individually. In addition, this regression included subject random intercepts and fixed effects of ratio and trial (both of which were z-scored before being entered into the regression), allowing us to aggregate information from the same baboon across experiments. The analysis reveals a large, positive, overall effect of one-to-one conditions having higher than chance accuracy (0 in this plot), 1–1 add begin: $b = 0.62$, OR = 1.87 [0.31, 0.94], $p < 0.001$, 1–1 add end: $b = 0.55$, OR = 1.74 [0.30, 0.78], $p < 0.001$, 1–1 subtract: $b = 0.69$, OR = 2.00 [0.38, 1.02], $p < 0.001$. Small, not statistically reliable effects of adding at the beginning or end were found. Across all data, baboons were also reliably above chance in the sequential condition, $b = 0.26$, OR = 1.29 [−0.04, 0.50], $p = 0.048$, but with lower accuracy than the one-to-one condition, $b = 0.60$, OR = 1.82 [0.39, 0.80], $p < 0.001$.

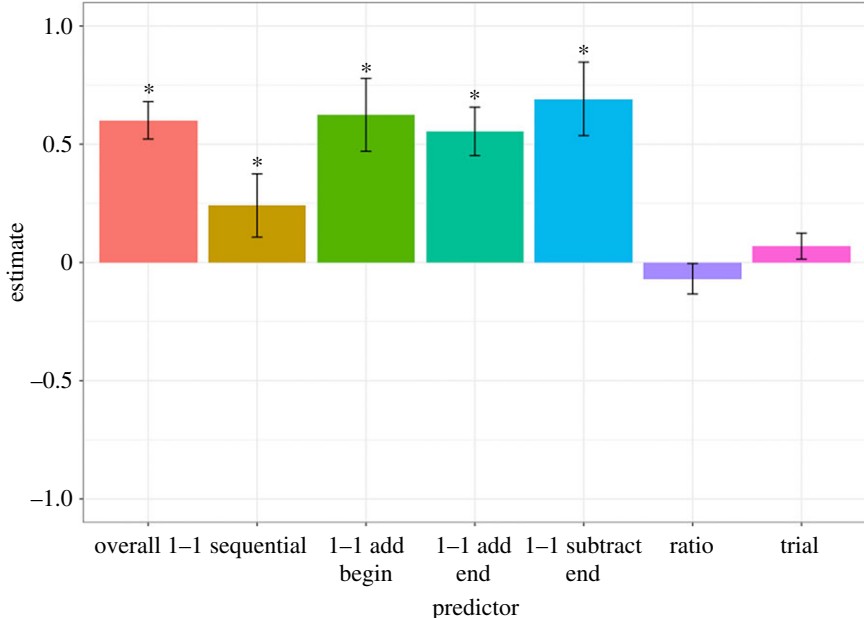

**Figure 6.** Coefficients from two overall logistic regressions across Experiments 1–3. The first regression compared overall one-to-one (red bar) and sequential (yellow bar) conditions. The second regression broke down the overall one-to-one condition by type of trial: extra added at the beginning of the trial (green bar), extra added at the end (teal bar) and subtraction (blue bar). Ratio (purple bar) and trial number (pink bar) were also included in the regressions, as was a random effect of subject. Chance-level accuracy corresponds to $y = 0$. Error bars are standard error of the estimate. *$p < 0.05$.

## 4. Discussion

When tasked with making quantity discriminations with difficult ratios, baboons were more accurate when the quantities were baited in a one-to-one fashion than when baited sequentially. That two of the three baboons failed in the sequential condition was unsurprising; indeed, it was part of our experimental design: to ensure that performance in the one-to-one baiting conditions could not be attributed to the ANS, we used test pairs that were not easily discriminable using the ANS. The critical result is that despite failing to discriminate these quantities when baited sequentially, these baboons successfully discriminated the same quantities when baited in a one-to-one fashion. This suggests that the baboons may have some understanding of one-to-one correspondence, over both addition and subtraction operations. Baboons could use the one-to-one baiting to infer equality between the sets to determine the point at which the sets became unequal and identify the set with extra items.

An alternative hypothesis for the baboons' success in the one-to-one addition condition is that they simply chose the cup that the experimenter interacted with last. We tested this hypothesis by adding a random cup tap after both cups were baited. There was no significant effect of tapping the correct cup in either experiment, indicating that baboons were not using a strategy of choosing the last cup the experimenter interacted with to do the task. Moreover, an examination of the relative effects of the last cup baited, the cup containing more food pieces and baiting condition on the cup chosen showed that baboons based their choice on the quantities contained by the cups and not the last cup baited.

We further explored the potential low-level strategy of choosing the last cup baited by testing the baboons on one-to-one addition trials where the extra food piece was added at the beginning of the trial, rather than at the end of the trial. Baboons' accuracy was significantly better than chance on both types of trials, suggesting that they were not relying on a 'choose the last cup baited' strategy. These results indicate that baboons were not relying on a strategy of choosing the last cup baited in this task.

We investigated whether performance differences in sequential and one-to-one trials could be explained by differences in baiting time, rather than an understanding of one-to-one correspondence. Sequential trials required more baiting events, and thus longer baiting time, than one-to-one trials, potentially leading to less accurate memory of the quantities in the sequential condition. We found no effect of baiting time on accuracy. This indicates that the lower accuracy in the sequential condition cannot be explained by the longer baiting time in that condition.

Contrary to what we would expect based on prior studies using sequential quantity discrimination tasks, we found no ratio effect (where discrimination accuracy decreases as the ratio between quantities increases) in the sequential condition. However, the range of ratios tested was narrow (0.75–0.83), especially for our sample size. When these baboons were previously tested on a wider range of quantity discriminations, their performance exhibited the ratio effect characteristic of the ANS [42].

Baboons were able to make finer numerical discriminations than predicted by their ANS in the one-to-one baiting conditions (addition and subtraction), indicating they have at least a rudimentary understanding of one-to-one correspondence and equality or exactness. One-to-one correspondence could be a form of logic that develops independently of language. Although the independence of cardinality and counting is debated in the human literature [14,23–30,43], our finding that baboons seem to use a one-to-one heuristic to evaluate the relative values of food sets supports the conclusion that they are independent. The mechanistic question that remains is how multiple mechanisms for judging quantity—the ANS, parallel individuation and logical rules like one-to-one correspondence—work together during economic and social decision-making to produce coherent behaviour. Additional work is also needed to ascertain whether this logical ability seen in baboons is a limited version of the one humans possess. For example, humans can impose one-to-one relations on sets by generating correspondence between them (e.g. muffins to muffin tins), whereas non-human primates are unlikely to generate sets at all. Non-human primates might need the environment to highlight correspondence, either temporally or spatially, in order to recognize a one-to-one relation between sets. Additionally, it is unclear how this logical rule is represented non-verbally, or which quantitative systems may make use of it in humans, whether ANS, parallel individuation or basic concepts of singularity and plurality [44]. Nonetheless, our findings indicate that one-to-one correspondence, a logical rule that requires intuitions about equality and a possible building block of exact counting, has a pre-linguistic origin.

Ethics. Research with the subjects in this study was approved by the Seneca Park Zoo Conservation & Research Committee and the University of Rochester IACUC (Protocol 2017-004).

Data accessibility. The data are available on a permanent third-party archive, Open Science Framework: https://osf.io/s6hck/?view_only=56dfc5c160c348cc875d4fcb959919f4.

Authors' contributions. S.E.K. and J.F.C. developed the study concept and designed the studies. S.E.K. and A.M.A. collected data and conducted analyses, under the advisement of J.F.C. and S.T.P. S.E.K. wrote the manuscript. All authors provided revisions and approved the final version of the manuscript for submission.

Competing interests. We declare we have no competing interests.

Funding. Funding was provided by the James S. McDonnell Foundation (grant no. 220020300), the Alfred P. Sloan Foundation (grant no. FG-BR2013-019), the National Science Foundation (Education Core Research Grant DRL-1459625, Division of Research on Learning grant no. 1760874 and Graduate Research Fellowship DGE-1419118) and the National Institutes of Health (NICHD grant no. 1R01HK085996).

Acknowledgements. Thanks to Steve Ferrigno, Katie Brown, Isabelle Boni, Gabrielle Bueno, Abigail Haslinger and Yinghui Qiu for assistance in data collection and video coding. Thanks to the Seneca Park Zoo for allowing us to work with their baboons, and to Jenna Bovee, Nicole McEvily and Jeb McConnell for research support.

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
