## [Reviewer comments · Royal Society Open Science]

Review History

RSOS-190495.R0 (Original submission)

Review form: Reviewer 1

Is the manuscript scientifically sound in its present form?

No

Are the interpretations and conclusions justified by the results?

No

Is the language acceptable?

Yes

Do you have any ethical concerns with this paper?

No

Have you any concerns about statistical analyses in this paper?

No

Recommendation?

Major revision is needed (please make suggestions in comments)

Comments to the Author(s)

Three Olive baboons were tested on a relative numerosness discrimination under several conditions: sequential (one cylinder baited some number of times, then the other cylinder baited some different number of times), simultaneous (both cylinders baited concurrently, but with a single cylinder receiving one additional item), and subtraction (as above, but with an item or two then removed from one cylinder to make it less than the other). After training, the animals were better with the simultaneous than the sequential tests, and better with the subtraction than the 'addition' condition. The authors conclude this as evidence for 1-to-1 correspondence – an important building block of counting, exact number, and equality – in nonhuman primates.

This is an interesting and well-written submission on an important topic. The pattern of results is clear: when you give monkeys a cue other than quantity on which to make judgments, performance is improved by this cue. The critical question for interpreting these results then becomes, "What is the cue that the animals used?" The authors conclude that the cue involved 1-to-1 correspondence, and reject the alternative interpretation (the animals simply go the the last cylinder baited and avoid the cylinder from which food is removed) because a control (tapping a cylinder) did not affect performance.

However, I'm not convinced by the report that tapping the cylinder is a suitable control for "go to the last baited well." The problem, of course, is that "go to the last baited well, and avoid any well from which food has been removed" is perfectly confounded with 1-to-1 correspondence, at least as operationalized here. The authors are probably satisfied that if the monkeys are responding to the last well in which food was individually placed, and doing so in the simultaneous but not the sequential condition, then this is ample evidence of 1-to-1 correspondence. But such a judgment based on the 'last in/last out' confound (which could have been acquired in the training trials, if indeed was not the baboons' natural biases) would not require anything like "understanding that two sets are equal if each item in one set corresponds to exactly one item in the second set" (Page 1.15). Indeed, their paradigm as used cannot show this at all. I would like to have seen trials like this

C1: b b b b b b b

C2: b b b b b b b

(where C means cylinder and b is a bating event) to show that performance was at 50%, and trials like this

C1: b b b b b b b

C2: b b b b b b b

Or

C1: b b b b

C2: b b b

to suggest that the animals were not simply using the confounding "last in/last out" strategy.

I'm curious: if the experimenters used this same procedure to test 0 versus 1 trials, which I predict the animals would pass at levels significantly greater than chance, would the authors conclude that the animals used 1-to-1 correspondence (i.e., both cylinders had matching 0 events prior)? If

1-to-1 isn't required to solve a 0-versus-1 trial in their paradigm, I don't think it is required to solve a 5-versus-6 trial in this paradigm.

In conclusion, I think the authors have reported the first two tests of a series of conditions that will be required ultimately to demonstrate "understanding that two sets are equal if each item in one set corresponds to exactly one item in the second set." Once they revise the manuscript with additional data/conditions that bolster their claim that cylinder-tapping is an adequate control for what I've called here the "last in/last out" response strategy, I believe it will make a substantive contribution to the literature.

Review form: Reviewer 2

Is the manuscript scientifically sound in its present form?

Yes

Are the interpretations and conclusions justified by the results?

No

Is the language acceptable?

Yes

Do you have any ethical concerns with this paper?

No

Have you any concerns about statistical analyses in this paper?

No

Recommendation?

Major revision is needed (please make suggestions in comments)

Comments to the Author(s)

Major comments:

This was a fascinating paper to read, and it's nice to see the further investigation of 1-to-1 correspondence recognition among non-human primates. The demonstration that some baboons understand 1-to-1 correspondence would be a critical finding, informing ongoing debates on the role that this ability plays in fostering numerical cognition, and on the role that language plays (or does not play) in facilitating 1-to-1 correspondence. This is particularly true since the ability of anumeric humans to recognize 1-to-1 correspondence for sets larger than 3-4 is debated. (In fact, the extent to which this capacity is exhibited in anumeric human populations is contested in ways not sufficiently acknowledged in this paper--a point returned to in the "minor comments" section below.)

I am not inherently skeptical of the general conclusion that baboons are capable of recognizing 1-to-1 correspondence. Nevertheless, in my view the data presented in this paper are explainable without appealing to such a capacity. Available alternate explanations of the two experiments' results seem more parsimonious at present. If there are legitimate reasons why the alternate explanations proposed below are untenable, these reasons need to be laid out in the paper. It is possible that the three baboons tested are indeed relying on 1-to-1 correspondence. But why would we assume so if simpler explanations exist? I do think there is a way to use the same

general approach to more clearly (though likely not conclusively) demonstrate that 1-to-1 correspondence is at work, but it will require re-running the experiments with higher quantities.

Consider again the methods used for each experiment: For experiment 1, the 1-to-1 addition task: “cylinders were baited simultaneously, one item at a time, with the baiting of the larger quantity continuing alone once the smaller quantity had been baited.” So the obvious question, then, is whether the three baboons are simply choosing (about $\frac{2}{3}$ of the time anyway) the cylinder that was most recently baited. This is a pretty simple explanation, requiring no quantity matching whatsoever. To control for this possibility, the authors state that one of the cylinders was randomly chosen and tapped, to ensure that the choices were not merely based on which cup the experimenter last interacted with. But if the baboons are fixated on a given treat and see it descend into a given cylinder, how exactly would tapping on a cylinder control for the recency of treat placement? Presumably a baboon can differentiate between tapping and the actual placement of a treat in a cylinder. So the performance can simply be explained if the baboons tend to pick the cylinder wherein they last saw a treat placed, just a few seconds prior.

Given that some anumeric human populations seem to struggle with 1-to-1 matching for larger sets (Spaepen et al. 2011 PNAS, Gordon 2004 Science, Everett & Madora 2012 Cognitive Science), here is one possible way around this issue: Run the task with larger sets as well. I would be more convinced of the suggested interpretation if, for instance, baboon performance deteriorated for set discrepancies like 7 vs. 8 treats. This would suggest that the baboons are not simply choosing the cylinder wherein the last treat was placed, but are matching quantities to the extent that they can. Without this or some other refinement of this approach, I’m not sure what these results actually demonstrate.

One could argue that the 1-to-1 subtraction experiment controls for any recency bias, because in this task the cylinder with more treats is not simply the one in which a treat was last placed. But it seems that the results for this experiment can also be explained without appealing to 1-to-1 matching: Perhaps Peperella and Sabina are simply avoiding the cylinder from which they just saw a treat removed. This may be cognitively more elaborate than the strategy used for the addition experiment, but it still seems simpler than assuming that their choices are based on the recognition of exact quantity correspondence. Also, only two of the three (already screened) subjects performed above chance for this task. This is consistent with the notion that this task is more difficult than that in experiment 1, and may not rely on the same fundamental principle or strategy (exact quantity recognition). Perhaps Peperella and Sabina have some idea that “treat removal is a bad thing”, maybe because they have seen treats removed from cylinders in the past. To be clear, I’m not claiming that this strategy is necessarily operative, merely that I do not find the interpretation offered in the paper compelling at present. I suspect many readers would likely reach the same conclusion.

As with the first experiment, I would find the authors’ interpretation of the subtraction experiment more convincing if performance deteriorated with higher quantities that were not tested. If so, we could be more confident that the responses associated with smaller sets actually relate to the baboons’ ability to keep track of exact correspondence across the cylinders, rather than relating to the recency of “treat movement” (and not cylinder tapping). As an extreme example, consider if 15 treats were placed in each cylinder, and then one treat was removed from a cylinder. If Peperella’s and Sabina’s selections became random, this would offer some evidence that they are not simply using an “avoid-a-cylinder-if-it-has-a-treat-removed” strategy. In short, for both experiments the strategies suggested in this review would be just as easy for larger sets. But the 1-to-1 correspondence strategy would likely see deteriorating results. (A case could be made that the deteriorating results would not occur for larger sets even if they’re really using 1-to-1 matching, but I’m trying to offer a way forward that would at least convince more readers.)

Also, I think the “avoidance” strategy more readily explains the fact that “Within the subtraction condition, the number of food pieces subtracted (1 or 2) did not affect accuracy.” If 1-to-1 correspondence recognition is operative, shouldn’t a +2 discrepancy across cylinders lead to a greater likelihood of choosing the cylinder with more treats?

Minor comments:

I think both the title and the abstract overreach. It’s not clear that these baboons are “understanding 1-to-1 correspondence”. “Preliminary evidence of 1-to-1 matching of small quantities in baboons” would more accurately convey the key findings, even assuming alternate interpretations are more clearly addressed.

Page 1, Line 48. Along with these three “cognitive systems”, one factor that is often unacknowledged is grammatical number. Some recent work suggests that grammatical dual markers, for example, help kids acquire exact number more rapidly. (Marusic et al. 2016, PLOS ONE)

Page 2, Line 20. With respect to “populations without exact number words”, this should be clarified with a line or two. Partial understanding of 1-to-1 correspondence has been demonstrated, but for two studies amongst the Piraha (Gordon 2004, Everett & Madora 2012), this has only been demonstrated for quantities < 4. For higher quantities, there was clear degradation in one-to-one matching abilities. This should be pointed out even though Frank et al. (2008), cited in this paper, obtained different results for higher quantities in a basic 1-to-1 matching task. There is apparently debate as to what motivated this cross-study discrepancy. Also, I’m surprised that Spaepen et al. (2011, PNAS) is not cited here. In that study, a numeric homesigners also struggled with exact recognition for larger sets.

Page 3, lines 42-49. “Establishing a capacity....requires just one example.” Fair enough, but it would still be nice to know how many baboons were required to get to a sample of 3. How many baboons could not reach the given criterion in the training phase?

Discussion of procedure and results: See major comment above.

Page 4, lines 22-23. Some baboons likely exhibit stronger food preferences than others. I’m assuming the same items were used for each baboon, across all trials and both experiments, once their preference was established. Still, it would be helpful to see this explicitly stated. “Depending on the preference of the subject” is ambiguous and could mean that, e.g., Pearl preferred several of the items, and perhaps the items varied for Pearl across tasks.

Page 6: Relatedly, the data for Pearl are difficult to interpret: Significantly above chance on the sequential condition, unlike Peperella and Sabina. But not significantly above chance in the 1-to-1 subtraction experiment, again unlike the other two. If these tasks are really getting at numerical cognition, why is Pearl so poor at the 1-to-1 subtraction task while excelling at the other tasks, including the ostensibly more difficult sequential task? I realize we can’t know exactly what’s happening with an individual baboon, but Pearl’s performance hints that different strategies may be in effect across the two experiments, and perhaps across baboons.

Page 8, line 46. I agree, the range of ratios tested here was quite narrow. Particularly given that these baboons previously exhibited ratio effects for a wider range, I think running the same experiment with different ratios and larger quantities would be helpful.

Page 8, line 53. Again, I’m not sure this conclusion is warranted.

Page 8, line 55, and Page 9, line 4. The claim that “the logic of exact equality develops independently of the concept of exact cardinality from verbal counting” is actually contestable, judging from some of the literature with anumeric adults.

Finally, let me reiterate how interesting I found this work. Despite the skepticism I have expressed towards the current interpretation of the results, I really hope this approach is pursued further.

Decision letter (RSOS-190495.R0)

03-Jul-2019

Dear Ms Koopman,

The editors assigned to your paper ("Understanding 1-to-1 Correspondence without Language") have now received comments from reviewers. We would like you to revise your paper in accordance with the referee and Associate Editor suggestions which can be found below (not including confidential reports to the Editor). Please note this decision does not guarantee eventual acceptance.

Please submit a copy of your revised paper before 26-Jul-2019. Please note that the revision deadline will expire at 00.00am on this date. If we do not hear from you within this time then it will be assumed that the paper has been withdrawn. In exceptional circumstances, extensions may be possible if agreed with the Editorial Office in advance. We do not allow multiple rounds of revision so we urge you to make every effort to fully address all of the comments at this stage. If deemed necessary by the Editors, your manuscript will be sent back to one or more of the original reviewers for assessment. If the original reviewers are not available, we may invite new reviewers.

- Data accessibility

If you wish to submit your supporting data or code to Dryad (<http://datadryad.org/>), or modify your current submission to dryad, please use the following link:
<http://datadryad.org/submit?journalID=RSOS&manu=RSOS-190495>

- Competing interests

- Authors' contributions

- Acknowledgements

- Funding statement

on behalf of Dr Mark Walton (Associate Editor) and Essi Viding (Subject Editor)
 openscience@royalsociety.org

Comments to Author:

Reviewers' Comments to Author:

Reviewer: 1

Comments to the Author(s)

Three Olive baboons were tested on a relative numerosness discrimination under several conditions: sequential (one cylinder baited some number of times, then the other cylinder baited some different number of times), simultaneous (both cylinders baited concurrently, but with a single cylinder receiving one additional item), and subtraction (as above, but with an item or two then removed from one cylinder to make it less than the other). After training, the animals were better with the simultaneous than the sequential tests, and better with the subtraction than the 'addition' condition. The authors conclude this as evidence for 1-to-1 correspondence – an important building block of counting, exact number, and equality – in nonhuman primates.

This is an interesting and well-written submission on an important topic. The pattern of results is clear: when you give monkeys a cue other than quantity on which to make judgments, performance is improved by this cue. The critical question for interpreting these results then becomes, "What is the cue that the animals used?" The authors conclude that the cue involved 1-to-1 correspondence, and reject the alternative interpretation (the animals simply go the the last cylinder baited and avoid the cylinder from which food is removed) because a control (tapping a cylinder) did not affect performance.

However, I'm not convinced by the report that tapping the cylinder is a suitable control for "go to the last baited well." The problem, of course, is that "go to the last baited well, and avoid any well from which food has been removed" is perfectly confounded with 1-to-1 correspondence, at least as operationalized here. The authors are probably satisfied that if the monkeys are responding to the last well in which food was individually placed, and doing so in the simultaneous but not the sequential condition, then this is ample evidence of 1-to-1 correspondence. But such a judgment based on the 'last in/last out' confound (which could have been acquired in the training trials, if indeed was not the baboons' natural biases) would not require anything like "understanding that two sets are equal if each item in one set corresponds to exactly one item in the second set" (Page 1.15). Indeed, their paradigm as used cannot show this at all. I would like to have seen trials like this

C1: b b b b b b b

C2: b b b b b b b

(where C means cylinder and b is a bating event) to show that performance was at 50%, and trials like this

C1: b b b b b b b

C2: b b b b b b b

Or

C1: b b b b

C2: b b b

to suggest that the animals were not simply using the confounding “last in/last out” strategy.

I’m curious: if the experimenters used this same procedure to test 0 versus 1 trials, which I predict the animals would pass at levels significantly greater than chance, would the authors conclude that the animals used 1-to-1 correspondence (i.e., both cylinders had matching 0 events prior)? If 1-to-1 isn’t required to solve a 0-versus-1 trial in their paradigm, I don’t think it is required to solve a 5-versus-6 trial in this paradigm.

In conclusion, I think the authors have reported the first two tests of a series of conditions that will be required ultimately to demonstrate “understanding that two sets are equal if each item in one set corresponds to exactly one item in the second set.” Once they revise the manuscript with additional data/conditions that bolster their claim that cylinder-tapping is an adequate control for what I’ve called here the “last in/last out” response strategy, I believe it will make a substantive contribution to the literature.

Reviewer: 2

Comments to the Author(s)

Major comments:

This was a fascinating paper to read, and it’s nice to see the further investigation of 1-to-1 correspondence recognition among non-human primates. The demonstration that some baboons understand 1-to-1 correspondence would be a critical finding, informing ongoing debates on the role that this ability plays in fostering numerical cognition, and on the role that language plays (or does not play) in facilitating 1-to-1 correspondence. This is particularly true since the ability of anumeric humans to recognize 1-to-1 correspondence for sets larger than 3-4 is debated. (In fact, the extent to which this capacity is exhibited in anumeric human populations is contested in ways not sufficiently acknowledged in this paper--a point returned to in the “minor comments” section below.)

I am not inherently skeptical of the general conclusion that baboons are capable of recognizing 1-to-1 correspondence. Nevertheless, in my view the data presented in this paper are explainable without appealing to such a capacity. Available alternate explanations of the two experiments’ results seem more parsimonious at present. If there are legitimate reasons why the alternate explanations proposed below are untenable, these reasons need to be laid out in the paper. It is possible that the three baboons tested are indeed relying on 1-to-1 correspondence. But why would we assume so if simpler explanations exist? I do think there is a way to use the same general approach to more clearly (though likely not conclusively) demonstrate that 1-to-1 correspondence is at work, but it will require re-running the experiments with higher quantities.

Consider again the methods used for each experiment: For experiment 1, the 1-to-1 addition task: “cylinders were baited simultaneously, one item at a time, with the baiting of the larger quantity continuing alone once the smaller quantity had been baited.” So the obvious question, then, is whether the three baboons are simply choosing (about $\frac{2}{3}$ of the time anyway) the cylinder that was most recently baited. This is a pretty simple explanation, requiring no quantity matching whatsoever. To control for this possibility, the authors state that one of the cylinders was randomly chosen and tapped, to ensure that the choices were not merely based on which cup the experimenter last interacted with. But if the baboons are fixated on a given treat and see it descend into a given cylinder, how exactly would tapping on a cylinder control for the recency of treat placement? Presumably a baboon can differentiate between tapping and the actual

placement of a treat in a cylinder. So the performance can simply be explained if the baboons tend to pick the cylinder wherein they last saw a treat placed, just a few seconds prior.

Given that some anumeric human populations seem to struggle with 1-to-1 matching for larger sets (Spaepen et al. 2011 PNAS, Gordon 2004 Science, Everett & Madora 2012 Cognitive Science), here is one possible way around this issue: Run the task with larger sets as well. I would be more convinced of the suggested interpretation if, for instance, baboon performance deteriorated for set discrepancies like 7 vs. 8 treats. This would suggest that the baboons are not simply choosing the cylinder wherein the last treat was placed, but are matching quantities to the extent that they can. Without this or some other refinement of this approach, I'm not sure what these results actually demonstrate.

One could argue that the 1-to-1 subtraction experiment controls for any recency bias, because in this task the cylinder with more treats is not simply the one in which a treat was last placed. But it seems that the results for this experiment can also be explained without appealing to 1-to-1 matching: Perhaps Peperella and Sabina are simply avoiding the cylinder from which they just saw a treat removed. This may be cognitively more elaborate than the strategy used for the addition experiment, but it still seems simpler than assuming that their choices are based on the recognition of exact quantity correspondence. Also, only two of the three (already screened) subjects performed above chance for this task. This is consistent with the notion that this task is more difficult than that in experiment 1, and may not rely on the same fundamental principle or strategy (exact quantity recognition). Perhaps Peperella and Sabina have some idea that "treat removal is a bad thing", maybe because they have seen treats removed from cylinders in the past. To be clear, I'm not claiming that this strategy is necessarily operative, merely that I do not find the interpretation offered in the paper compelling at present. I suspect many readers would likely reach the same conclusion.

As with the first experiment, I would find the authors' interpretation of the subtraction experiment more convincing if performance deteriorated with higher quantities that were not tested. If so, we could be more confident that the responses associated with smaller sets actually relate to the baboons' ability to keep track of exact correspondence across the cylinders, rather than relating to the recency of "treat movement" (and not cylinder tapping). As an extreme example, consider if 15 treats were placed in each cylinder, and then one treat was removed from a cylinder. If Peperella's and Sabina's selections became random, this would offer some evidence that they are not simply using an "avoid-a-cylinder-if-it-has-a-treat-removed" strategy. In short, for both experiments the strategies suggested in this review would be just as easy for larger sets. But the 1-to-1 correspondence strategy would likely see deteriorating results. (A case could be made that the deteriorating results would not occur for larger sets even if they're really using 1-to-1 matching, but I'm trying to offer a way forward that would at least convince more readers.)

Also, I think the "avoidance" strategy more readily explains the fact that "Within the subtraction condition, the number of food pieces subtracted (1 or 2) did not affect accuracy." If 1-to-1 correspondence recognition is operative, shouldn't a +2 discrepancy across cylinders lead to a greater likelihood of choosing the cylinder with more treats?

Minor comments:

I think both the title and the abstract overreach. It's not clear that these baboons are "understanding 1-to-1 correspondence". "Preliminary evidence of 1-to-1 matching of small quantities in baboons" would more accurately convey the key findings, even assuming alternate interpretations are more clearly addressed.

Page 1, Line 48. Along with these three “cognitive systems”, one factor that is often unacknowledged is grammatical number. Some recent work suggests that grammatical dual markers, for example, help kids acquire exact number more rapidly. (Marusic et al. 2016, PLOS ONE)

Page 2, Line 20. With respect to “populations without exact number words”, this should be clarified with a line or two. Partial understanding of 1-to-1 correspondence has been demonstrated, but for two studies amongst the Piraha (Gordon 2004, Everett & Madora 2012), this has only been demonstrated for quantities < 4. For higher quantities, there was clear degradation in one-to-one matching abilities. This should be pointed out even though Frank et al. (2008), cited in this paper, obtained different results for higher quantities in a basic 1-to-1 matching task. There is apparently debate as to what motivated this cross-study discrepancy. Also, I’m surprised that Spaepen et al. (2011, PNAS) is not cited here. In that study, anumeric homesigners also struggled with exact recognition for larger sets.

Page 3, lines 42-49. “Establishing a capacity....requires just one example.” Fair enough, but it would still be nice to know how many baboons were required to get to a sample of 3. How many baboons could not reach the given criterion in the training phase?

Discussion of procedure and results: See major comment above.

Page 4, lines 22-23. Some baboons likely exhibit stronger food preferences than others. I’m assuming the same items were used for each baboon, across all trials and both experiments, once their preference was established. Still, it would be helpful to see this explicitly stated. “Depending on the preference of the subject” is ambiguous and could mean that, e.g., Pearl preferred several of the items, and perhaps the items varied for Pearl across tasks.

Page 6: Relatedly, the data for Pearl are difficult to interpret: Significantly above chance on the sequential condition, unlike Peperella and Sabina. But not significantly above chance in the 1-to-1 subtraction experiment, again unlike the other two. If these tasks are really getting at numerical cognition, why is Pearl so poor at the 1-to-1 subtraction task while excelling at the other tasks, including the ostensibly more difficult sequential task? I realize we can’t know exactly what’s happening with an individual baboon, but Pearl’s performance hints that different strategies may be in effect across the two experiments, and perhaps across baboons.

Page 8, line 46. I agree, the range of ratios tested here was quite narrow. Particularly given that these baboons previously exhibited ratio effects for a wider range, I think running the same experiment with different ratios and larger quantities would be helpful.

Page 8, line 53. Again, I’m not sure this conclusion is warranted.

Page 8, line 55, and Page 9, line 4. The claim that “the logic of exact equality develops independently of the concept of exact cardinality from verbal counting” is actually contestable, judging from some of the literature with anumeric adults.

Finally, let me reiterate how interesting I found this work. Despite the skepticism I have expressed towards the current interpretation of the results, I really hope this approach is pursued further.

Author's Response to Decision Letter for (RSOS-190495.R0)

See Appendix A.

RSOS-190495.R1 (Revision)

Review form: Reviewer 1

Is the manuscript scientifically sound in its present form?

Yes

Are the interpretations and conclusions justified by the results?

Yes

Is the language acceptable?

Yes

Do you have any ethical concerns with this paper?

No

Have you any concerns about statistical analyses in this paper?

No

Recommendation?

Accept as is

Comments to the Author(s)

I commend the authors for thoughtful consideration of the initial reviews. The additional data and analysis strengthen the authors' conclusions, and make the findings much more compelling. I have no additional concerns about the submission, and think it will interest many scholars who study topics related to the target question.

Review form: Reviewer 2

Is the manuscript scientifically sound in its present form?

Yes

Are the interpretations and conclusions justified by the results?

No

Is the language acceptable?

Yes

Do you have any ethical concerns with this paper?

No

Have you any concerns about statistical analyses in this paper?

No

Recommendation?

Accept with minor revision (please list in comments)

Comments to the Author(s)

This version of the manuscript represents an improvement, and I like the approach the authors have adopted to address the principal concern expressed by both myself and the other reviewer. The additional experiment and the new random effects regression analysis make a more compelling case that 1-to-1 correspondence is at work. Though these results do still suggest that the expressed concern is relevant to the general experimental design, and also do not rule out the possibility that the baboons relied on the ANS.

Let me stress that I think this paper makes a very nice contribution to the literature on primate quantitative cognition. Yet I'm still not sure just how much it impacts the ongoing debates on the role(s) language plays in 1-to-1 correspondence. Consider this new line offered in the paper's conclusion, which is an improvement on the claim it replaced but still, to me, problematic: "Thus although the independence of cardinality and counting is debated in the human literature, our finding that baboons seem to use a 1-to-1 heuristic to evaluate the relative values of food sets supports the conclusion that they are independent." Here's the issue: All of the work with non-counting humans finds that they "seem to use a 1-to-1 heuristic" for smaller quantities, but in some key results there is a gradual decline thereafter consistent with ANS reliance, even in some tasks where we'd expect lossless 1-1 matching. To show that the ANS isn't being engaged to help track quantities in these trials, even in non-sequential tasks, it would be nice to see comparable ranges tested to show whether there is a decline analogous to that previously observed with non-counters. While it appears there is not, for this study we are stuck somewhere in the middle of the relevant range and so it is hard to evaluate. Particularly since the baboons are so much closer to chance than ceiling, especially in the case of the trials contrasting 5 vs. 6, no straightforward interpretation yet presents itself. The authors acknowledge that the ratios tested represent a narrow range. It is this narrow range that makes the relevance of this work to language debates difficult to evaluate at present.

They note in their response that "1-to-1 correspondence is not limited by magnitude; it involves the pairing of items across two sets, not the enumeration or tracking of entire quantities. Theoretically, we would therefore expect performance on 7-versus-8 to be similar to performance on 17-versus-18 or 77-versus-78, etc." True, but attempts to pair items in sets may involve ANS usage, absent language, in actual practice. That's the question, do non-counters rely on estimation even when we might expect they would not on theoretical grounds? The precipitous decline around 4 or 5 seems to suggest ANS-reliance in some studies. The critical point here is that we have to test a sufficient range of magnitudes to even see if the signatures of the ANS are absent. Otherwise that interpretation will be available. And if we just assume 1-to-1 tracking will hold and is by definition immune to magnitude effects, why even test any range of quantities at all? In the present study we can't contrast the results for, say, 2 vs. 3 with 6 vs. 7 to see if there is a similar drop off as what has been observed with human non-counters. The ANS could still be playing a role when the stimuli presentation is non-sequential, if the baboons are keeping rough track of which bait has more food. We would expect decline across quantities if this were the case, and we do in fact get decline (see below) around the point some might predict. So even though the overall regression finds no significant ratio effect, the results could still be interpreted as being consistent with ANS usage.

Later on in their reply, the authors state that “a decrease in performance for larger quantities likely would indicate to us a loss of attention or motivation, that 1-to-1 correspondence was not used.” That’s an understandable interpretation, but using it as a basis to rule out tests on higher quantities runs the risk of unfalsifiability. How was the choice to stop at the 5 v. 6 contrast, and to test such a restricted range of ratios, motivated? (If it's due to something like the time available with the baboons, that's understandable, but perhaps just acknowledge further work is needed.) The authors state that “they used test pairs that were not easily discriminable using the ANS.” True, but these contrasts are still generally discriminable, though with high error rates, in research with non-counters. For that reason I was most interested in the 5 vs. 6 contrast. I took a look at the authors’ raw data, focusing on the results for that contrast (coded as 0.833 ratio) in the 3 experiments. And it is the only contrast with accuracy less than 60% in their results. In experiment 3 accuracy for this contrast is at 58%. This represents about an 8% drop from the 4 vs. 5 contrast in the same experiment--the same percentage by which the 5 vs. 6 results exceed chance. Regardless of the overall regression results, this raises questions to someone particularly interested in that contrast. Are the baboons already growing unmotivated? Would the deterioration continue, consistent with analogue estimates? We obviously wouldn’t need to get to “17 vs. 18 or 77 vs. 78” to get satisfactory answers to such questions. In fact, 6 vs. 7 and 7 vs. 8 would be very telling, one way or the other. The authors could reasonably counter that the baboons’ performance performance in experiment 1, in which the addition trials went significantly better than the sequential trials, shows that the ANS is not at work. Yet in trials for the sequential task focused on the 5 vs. 6 contrast, accuracy was 56%. This is nearly identical to the 58% accuracy for the 5 vs. 6 contrast in the critical third experiment. In the former case we are to conclude that the ANS was used, and in the latter case with nearly identical results, that it was not? One compelling point for the authors’ case is the performance on the subtraction task, since in that case performance on the 5 vs. 6 contrast was better.

Again, to be clear, I think these results are very intriguing! I just think the current claims related to language are too strong. As I noted in the first review, I was not inherently skeptical that baboons could exhibit 1-to-1 correspondence, I was just not convinced given the clear confounds. (The new logistic regression for experiment 1 factors in the "last-baited" confound, but note that “no main effects of the last cup baited” holds only barely (p.0.07).) This revision does point more clearly to 1-to-1 correspondence, but not as clearly as the authors suggest. Perhaps a better title would be “Signs of 1-1 correspondence in baboons”?

In any case, I look forward to seeing this work published. And, as noted at the outset of these comments, I think the authors’ approach to handling the main methodological critique (raised in both reviews) is elegant and effective. I would like to have seen it used to test a greater range of set sizes. Hopefully that will be possible in future work.

Decision letter (RSOS-190495.R1)

18-Sep-2019

Dear Ms Koopman,

On behalf of the Editors, I am pleased to inform you that your Manuscript RSOS-190495.R1 entitled "1-to-1 Correspondence without Language" has been accepted for publication in Royal Society Open Science subject to minor revision in accordance with the referee suggestions. Please find the referees' comments at the end of this email.

The reviewers and Subject Editor have recommended publication, but also suggest some minor

revisions to your manuscript. Therefore, I invite you to respond to the comments and revise your manuscript.

- Ethics statement

- Data accessibility

If you wish to submit your supporting data or code to Dryad (<http://datadryad.org/>), or modify your current submission to dryad, please use the following link:
<http://datadryad.org/submit?journalID=RSOS&manu=RSOS-190495.R1>

- Competing interests

- Authors' contributions

- Acknowledgements

- Funding statement

Please note that we cannot publish your manuscript without these end statements included. We

have included a screenshot example of the end statements for reference. If you feel that a given heading is not relevant to your paper, please nevertheless include the heading and explicitly state that it is not relevant to your work.

Because the schedule for publication is very tight, it is a condition of publication that you submit the revised version of your manuscript before 27-Sep-2019. Please note that the revision deadline will expire at 00.00am on this date. If you do not think you will be able to meet this date please let me know immediately.

Kind regards,
Lianne Parkhouse
Royal Society Open Science
openscience@royalsociety.org

on behalf of Dr Mark Walton (Associate Editor) and Essi Viding (Subject Editor)
 openscience@royalsociety.org

Reviewer comments to Author:

Reviewer: 1

I commend the authors for thoughtful consideration of the initial reviews. The additional data and analysis strengthen the authors' conclusions, and make the findings much more compelling. I have no additional concerns about the submission, and think it will interest many scholars who study topics related to the target question.

Reviewer: 2

This version of the manuscript represents an improvement, and I like the approach the authors have adopted to address the principal concern expressed by both myself and the other reviewer. The additional experiment and the new random effects regression analysis make a more compelling case that 1-to-1 correspondence is at work. Though these results do still suggest that the expressed concern is relevant to the general experimental design, and also do not rule out the possibility that the baboons relied on the ANS.

Let me stress that I think this paper makes a very nice contribution to the literature on primate quantitative cognition. Yet I'm still not sure just how much it impacts the ongoing debates on the role(s) language plays in 1-to-1 correspondence. Consider this new line offered in the paper's conclusion, which is an improvement on the claim it replaced but still, to me, problematic: "Thus although the independence of cardinality and counting is debated in the human literature, our finding that baboons seem to use a 1-to-1 heuristic to evaluate the relative values of food sets supports the conclusion that they are independent." Here's the issue: All of the work with non-counting humans finds that they "seem to use a 1-to-1 heuristic" for smaller quantities, but in some key results there is a gradual decline thereafter consistent with ANS reliance, even in some tasks where we'd expect lossless 1-1 matching. To show that the ANS isn't being engaged to help track quantities in these trials, even in non-sequential tasks, it would be nice to see comparable ranges tested to show whether there is a decline analogous to that previously observed with non-counters. While it appears there is not, for this study we are stuck somewhere in the middle of the relevant range and so it is hard to evaluate. Particularly since the baboons are so much closer to chance than ceiling, especially in the case of the trials contrasting 5 vs. 6, no straightforward interpretation yet presents itself. The authors acknowledge that the ratios tested represent a narrow range. It is this narrow range that makes the relevance of this work to language debates difficult to evaluate at present.

They note in their response that "1-to-1 correspondence is not limited by magnitude; it involves the pairing of items across two sets, not the enumeration or tracking of entire quantities. Theoretically, we would therefore expect performance on 7-versus-8 to be similar to performance on 17-versus-18 or 77-versus-78, etc." True, but attempts to pair items in sets may involve ANS usage, absent language, in actual practice. That's the question, do non-counters rely on estimation even when we might expect they would not on theoretical grounds? The precipitous decline around 4 or 5 seems to suggest ANS-reliance in some studies. The critical point here is that we have to test a sufficient range of magnitudes to even see if the signatures of the ANS are absent. Otherwise that interpretation will be available. And if we just assume 1-to-1 tracking will hold

and is by definition immune to magnitude effects, why even test any range of quantities at all? In the present study we can't contrast the results for, say, 2 vs. 3 with 6 vs. 7 to see if there is a similar drop off as what has been observed with human non-counters. The ANS could still be playing a role when the stimuli presentation is non-sequential, if the baboons are keeping rough track of which bait has more food. We would expect decline across quantities if this were the case, and we do in fact get decline (see below) around the point some might predict. So even though the overall regression finds no significant ratio effect, the results could still be interpreted as being consistent with ANS usage.

Later on in their reply, the authors state that "a decrease in performance for larger quantities likely would indicate to us a loss of attention or motivation, that 1-to-1 correspondence was not used." That's an understandable interpretation, but using it as a basis to rule out tests on higher quantities runs the risk of unfalsifiability. How was the choice to stop at the 5 v. 6 contrast, and to test such a restricted range of ratios, motivated? (If it's due to something like the time available with the baboons, that's understandable, but perhaps just acknowledge further work is needed.) The authors state that "they used test pairs that were not easily discriminable using the ANS." True, but these contrasts are still generally discriminable, though with high error rates, in research with non-counters. For that reason I was most interested in the 5 vs. 6 contrast. I took a look at the authors' raw data, focusing on the results for that contrast (coded as 0.833 ratio) in the 3 experiments. And it is the only contrast with accuracy less than 60% in their results. In experiment 3 accuracy for this contrast is at 58%. This represents about an 8% drop from the 4 vs. 5 contrast in the same experiment--the same percentage by which the 5 vs. 6 results exceed chance. Regardless of the overall regression results, this raises questions to someone particularly interested in that contrast. Are the baboons already growing unmotivated? Would the deterioration continue, consistent with analogue estimates? We obviously wouldn't need to get to "17 vs. 18 or 77 vs. 78" to get satisfactory answers to such questions. In fact, 6 vs. 7 and 7 vs. 8 would be very telling, one way or the other. The authors could reasonably counter that the baboons' performance performance in experiment 1, in which the addition trials went significantly better than the sequential trials, shows that the ANS is not at work. Yet in trials for the sequential task focused on the 5 vs. 6 contrast, accuracy was 56%. This is nearly identical to the 58% accuracy for the 5 vs. 6 contrast in the critical third experiment. In the former case we are to conclude that the ANS was used, and in the latter case with nearly identical results, that it was not? One compelling point for the authors' case is the performance on the subtraction task, since in that case performance on the 5 vs. 6 contrast was better.

Again, to be clear, I think these results are very intriguing! I just think the current claims related to language are too strong. As I noted in the first review, I was not inherently skeptical that baboons could exhibit 1-to-1 correspondence, I was just not convinced given the clear confounds. (The new logistic regression for experiment 1 factors in the "last-baited" confound, but note that "no main effects of the last cup baited" holds only barely (p.0.07).) This revision does point more clearly to 1-to-1 correspondence, but not as clearly as the authors suggest. Perhaps a better title would be "Signs of 1-1 correspondence in baboons"?

In any case, I look forward to seeing this work published. And, as noted at the outset of these comments, I think the authors' approach to handling the main methodological critique (raised in both reviews) is elegant and effective. I would like to have seen it used to test a greater range of set sizes. Hopefully that will be possible in future work.

Author's Response to Decision Letter for (RSOS-190495.R1)

See Appendix B.

Decision letter (RSOS-190495.R2)

01-Oct-2019

Dear Ms Koopman,

I am pleased to inform you that your manuscript entitled "1-to-1 Correspondence without Language" is now accepted for publication in Royal Society Open Science.

on behalf of Dr Mark Walton (Associate Editor) and Essi Viding (Subject Editor)
openscience@royalsociety.org

Appendix A

Sarah Koopman

Department of Brain & Cognitive Sciences
University of Rochester
Box 270268, Meliora Hall
Rochester, NY 14627
skoopman@ur.rochester.edu

August 15, 2019

Dear Editor:

Please consider our revision of manuscript “One-to-One Correspondence without Language” for publication in *Royal Society Open Science*.

The paper is the first investigation of the 1-to-1 correspondence principle in a non-human primate. This logical principle is thought to be important in counting and representing exact number, as it is the understanding that two sets are equal if each item in one set corresponds to exactly one item in the second set. In this study, baboons were given a quantity discrimination task involving two caches of food. In one condition, the quantities were baited in a manner requiring baboons to track the approximate quantities baited. In another condition, the quantities were baited in a manner that highlighted the 1-to-1 relation between the quantities. Baboons performed significantly better when 1-to-1 correspondence cues were provided, suggesting that they have at least a partial understanding of 1-to-1 correspondence. These results indicate that 1-to-1 correspondence, a logical rule that requires intuitions about equality and a possible building block of counting, has a primitive, pre-linguistic origin. This work provides new insight into the still-unclear process that children go through when learning to count and represent exact number.

The Reviewers made positive comments about the manuscript. Reviewer #1 commented that “this is an interesting and well-written submission on an important topic.” Reviewer #2 said “this was a fascinating paper to read.”

The Reviewers also made several criticisms and suggestions. We took this feedback seriously in our revision of the manuscript. Our revisions are detailed below and **highlighted** in the manuscript.

Reviewer #1

1. “...I’m not convinced by the report that tapping the cylinder is a suitable control for “got to the last baited well.” The problem, of course, is that “go to the last baited well, and avoid any well from which food has been removed” is perfectly confounded with 1-to-1 correspondence, at least as operationalized here. The authors are probably satisfied that if the monkeys are responding to the last well in which food was individually placed, and doing so in the simultaneous but not the sequential condition, then this is ample evidence of 1-to-1 correspondence. But such a judgment based on the “last in/last out” confound (which could have been acquired in the training trials, if indeed was not the baboons’ natural biases) would not require anything like “understanding that two sets are

equal if each item in one set corresponds to exactly one item in the second set” (Page 1.15). Indeed, their paradigm as used cannot show this at all. I would like to have seen trials like this

C1: b b b b b b b

C2: b b b b b b b

(where C means cylinder and b is a baiting event) to show that performance was at 50%, and trials like this

C1: b b b b b b b

C2: b b b b b b

Or

C1: b b b b

C2: b b b

to suggest that the animals were not simply using the confounding “last in/last out” strategy.”

Thank you for raising this concern and suggesting ways to address it. We have added an additional experiment that corresponds to the second suggestion, where the additional food piece was added at the beginning or end of the trial. We found that the baboons performed above chance in both conditions and that there was no significant difference in performance between the two conditions, suggesting that the baboons were not using a “last in/last out” strategy. This experiment now appears in the paper as Experiment 3, the methods of which are reported on pages 4-6 and results of which are reported on page 9. These results are discussed on page 10. A new random effects regression analysis across data from all three experiments also supports our conclusions (pages 9-10; see figure below).

We also conducted a new regression on Experiment 1 data (page 7), which examines the relative effects of the last cup baited, the cup containing more food

pieces, and baiting condition on the cup chosen by the baboons. The regression showed that baboons' choice was based on which cup contained more food pieces, but not the last cup baited or condition. This supports our conclusion that baboons did not just rely on a low-level strategy of choosing the last cup baited in the 1-to-1 condition.

Following the reviewer's suggestion, we have added new regression analyses which support our conclusions, but we are unable to conduct additional experiments (such as suggestions 1 and 3) since we no longer have access to the baboons.

2. "I'm curious: if the experimenters used this same procedure to test 0 versus 1 trials, which I predict the animals would pass at levels significantly greater than chance, would the authors conclude that the animals used 1-to-1 correspondence (i.e., both cylinders had matching 0 events prior)? If 1-to-1 isn't required to solve a 0-versus-1 trial in their paradigm, I don't think it is required to solve a 5-versus-6 trial in this paradigm."

Thank you for this question. In our view, there are multiple strategies that could be used to solve a 0-versus-1 trial, including tracking and comparing the one food piece using the parallel individuation system, comparing "something" to "nothing", and perhaps even 1-to-1 correspondence (although it seems unlikely that two empty cylinders are represented as "having 0 matching events"). However, a 5-versus-6 trial cannot be solved using the parallel individuation system (which can only track up to 3-4 objects) or a simple "something" versus "nothing" strategy; the baboons must somehow represent that the two cylinders are being baited with equal quantities of food until the extra food piece is added or subtracted. We contend that a 0-versus-1 trial is a different problem than a 5-versus-6 trial, and argue that it does not necessarily require the same cognitive processes. We agree that the precise nature of the animals' quantitative representations and mechanisms for solving this task requires further study in children and animals: "Additional work is needed to ascertain whether this logical ability seen in baboons is a limited version of the one humans possess. For example, humans can impose 1-to-1 relations on sets by generating correspondence between them (e.g., muffins to muffin tins) whereas non-human primates are unlikely to generate sets at all. Non-human primates might need the environment to highlight correspondence, either temporally or spatially, in order to recognize a 1-to-1 relation between sets" (page 11).

Reviewer 2

1. "...For experiment 1, the 1-to-1 addition task: "cylinders were baited simultaneously, one item at a time, with the baiting of the larger quantity continuing alone once the smaller quantity had been baited." So the obvious question, then, is whether the three baboons are simply choosing (about 2/3 of the time anyway) the cylinder that was most recently baited. This is a pretty simple explanation, requiring no quantity matching whatsoever. To control for this possibility, the authors state that one of the cylinders was randomly chosen and tapped, to ensure that the choices were not merely based on which

cup the experimenter last interacted with. But if the baboons are fixated on a given treat and see it descend into a given cylinder, how exactly would tapping on a cylinder control for the recency of treat placement? Presumably a baboon can differentiate between tapping and the actual placement of a treat in a cylinder. So the performance can simply be explained if the baboons tend to pick the cylinder wherein they last saw a treat placed, just a few seconds prior.”

Thank you for raising this concern. As explained above in response to Reviewer 1, we have added an additional experiment where the additional food piece was added at the beginning or end of the trial. We found that the baboons performed above chance in both conditions and that there was no significant difference in performance between the two conditions, suggesting that the baboons were not using a strategy based on the most recently baited cylinder. This experiment now appears in the paper as Experiment 3, the methods of which are reported on pages 4-6 and results of which are reported on page 9. These new data follow the reviewer’s suggestion, the results show that the baboons used quantity (not last cup baited in the task), and we have added a new random effects regression analysis incorporating these new data which supports our main conclusions, shown in Figure 6 (below). We also added analyses of Experiment 1 (page 7) which shows baboons used quantity and not last cup baited.

2. “Given that some anumeric human populations seem to struggle with 1-to-1 matching for larger sets (Spaepen et al. 2011 PNAS, Gordon 2004 Science, Everett & Madora 2012 Cognitive Science), here is one possible way around this issue: Run the task with larger sets as well. I would be more convinced of the suggested interpretation if, for instance, baboon performance deteriorated for set discrepancies like 7 vs. 8 treats. This would

suggest that the baboons are not simply choosing the cylinder wherein the last treat was placed, but are matching quantities to the extent that they can. Without this or some other refinement of this approach, I'm not sure what these results actually demonstrate.”

Thank you for this suggestion. However, we disagree that a deterioration in performance for larger quantities is indicative of the use of 1-to-1 correspondence. Indeed, the design of our studies is based on the widely-demonstrated ratio effect (where discrimination accuracy decreases as the ratio between quantities increases), and which we do NOT expect to see if 1-to-1 correspondence is being used to compare quantities. As explained in the Introduction (page 3), 1-to-1 correspondence is a logical rule based on equality and qualitative changes in the sets rather than a pure quantitative comparison. Thus, 1-to-1 correspondence is not limited by magnitude; it involves the pairing of items across two sets, not the enumeration or tracking of entire quantities. Theoretically, we would therefore expect performance on 7-versus-8 to be similar to performance on 17-versus-18 or 77-versus-78, etc. This is in contrast to quantity discrimination performance relying on the Approximate Number System (ANS), a behavioral signature of which is the ratio effect. Our strategy, therefore, was to compare performance on quantity discrimination trials that provided cues to 1-to-1 correspondence (1-to-1 addition and subtraction trials) and those that did not provide 1-to-1 cues, which could only be solved using the ANS (sequential trials).

3. “One could argue that the 1-to-1 subtraction experiment controls for any recency bias, because in this task the cylinder with more treats is not simply the one in which a treat was last placed. But it seems that the results for this experiment can also be explained without appealing to 1-to-1 matching: Perhaps Peperella and Sabina are simply avoiding the cylinder from which they just saw a treat removed. This may be cognitively more elaborate than the strategy used for the addition experiment, but it still seems simpler than assuming that their choices are based on the recognition of exact quantity correspondence. Also, only two of the three (already screened) subjects performed above chance for this task. This is consistent with the notion that this task is more difficult than that in experiment 1, and may not rely on the same fundamental principle or strategy (exact quantity recognition). Perhaps Peperella and Sabina have some idea that “treat removal is a bad thing”, maybe because they have seen treats removed from cylinders in the past. To be clear, I'm not claiming that this strategy is necessarily operative, merely that I do not find the interpretation offered in the paper compelling at present. I suspect many readers would likely reach the same conclusion.”

Thank you for raising this point. We have added a new random effects regression analysis that shows patterns across animals independently of Subject (see Figure 6 below). The results show that baboons succeeded across all 1-1 conditions in the study. In any experiment, it is always possible to come up with ad hoc strategies that could explain behavior. Our data demonstrate a general pattern across studies and manipulations, as confirmed by this new analysis across data from all three experiments (page 9-10), for which 1-to-1 correspondence, a simple logical rule that is not inherently linguistic, is a simple explanation.

4. “As with the first experiment, I would find the authors’ interpretation of the subtraction experiment more convincing if performance deteriorated with higher quantities that were not tested. If so, we could be more confident that the responses associated with smaller sets actually relate to the baboons’ ability to keep track of exact correspondence across the cylinders, rather than relating to the recency of “treat movement” (and not cylinder tapping). As an extreme example, consider if 15 treats were placed in each cylinder, and then one treat was removed from a cylinder. If Peperella’s and Sabina’s selections became random, this would offer some evidence that they are not simply using an “avoid-a-cylinder-if-it-has-a-treat-removed” strategy. In short, for both experiments the strategies suggested in this review would be just as easy for larger sets. But the 1-to-1 correspondence strategy would likely see deteriorating results. (A case could be made that the deteriorating results would not occur for larger sets even if they’re really using 1-to-1 matching, but I’m trying to offer a way forward that would at least convince more readers.)”

As explained in response to comment 2 above, we do not expect the use of 1-to-1 correspondence to lead to a decrease in performance for larger quantities, since that strategy does not require tracking the quantities baited. Rather, a decrease in performance for larger quantities likely would indicate to us a loss of attention or motivation, that 1-to-1 correspondence was not used. Use of larger quantities would yield ambiguous results.

5. “Also, I think the “avoidance” strategy more readily explains the fact that “Within the subtraction condition, the number of food pieces subtracted (1 or 2) did not affect accuracy.” If 1-to-1 correspondence recognition is operative, shouldn’t a +2 discrepancy across cylinders lead to a greater likelihood of choosing the cylinder with more treats?”

Consistent with this line of thought, we say in the Discussion that our interpretation is that the baboons use 1-1 correspondence to track equality between sets and then notice qualitative changes in the sets via additions or subtractions to identify the set with the larger amount: “Baboons could use the 1-to-1 baiting to infer equality between the sets to determine the point at which the sets became unequal and identify the set with extra items” (page 10). “Avoidance” isn’t an explanation of our data because one would have to explain how the baboon knows that subtraction matters in the context of this task where it is rewarded on every single trial regardless of choice. Moreover, baboons perform better in 1-1 addition than in sequential trials where the same values are presented also in an addition format – “avoidance” doesn’t explain that. Differences of +1 and +2 are not guaranteed to be large enough in accuracy to detect – the animals succeeded on both trial types. There could be many reasons for this null difference (which itself is not evidence of absence of difference) but the null outcome on that particular difference does not undermine the pattern of success we observed across 1-1 conditions that exceeds ANS performance.

6. “I think both the title and the abstract overreach. It’s not clear that these baboons are “understanding 1-to-1 correspondence.” “Preliminary evidence of 1-to-1 matching of small quantities in baboons” would more accurately convey the key findings, even assuming alternative interpretations are more clearly addressed.”

The results are not preliminary so we changed the title to “One-to-one correspondence without language” which is the topic of the paper.

7. “Page 1, Line 48. Along with these three “cognitive systems”, one factor that is often unacknowledged is grammatical number. Some recent work suggests that grammatical dual markers, for example, help kids acquire exact number more rapidly. (Marusic et al. 2016, PLOS ONE)”

Thank you for raising this point. The Introduction is distilling across many studies in the literature. We have added this citation to our discussion of how 1-to-1 correspondence may be involved in these cognitive systems in humans (page 11).

8. “Page 2, Line 20. With respect to “populations without exact number words”, this should be clarified with a line or two. Partial understanding of 1-to-1 correspondence has been demonstrated, but for two studies amongst the Piraha (Gordon 2004, Everett & Madora 2012), this has only been demonstrated for quantities < 4. For higher quantities, there was clear degradation in one-to-one matching abilities. This should be pointed out even though Frank et al. (2008), cited in this paper, obtained different results for higher quantities in a basic 1-to-1 matching task. There is apparently debate as to what motivated this cross-study discrepancy. Also, I’m surprised that Spaepen et al. (2011, PNAS) is not cited here. In that study, anumeric homesigners also struggled with exact recognition for larger sets.”

Thank you for this comment. We have added reference to this research to the Introduction (page 2), including citations to these three papers.

9. “Page 3, lines 42-49. “Establishing a capacity...requires just one example.” Fair enough, but it would still be nice to know how many baboons were required to get to a sample of 3. How many baboons could not reach the given criterion in the training phase?”

We attempted to test each of the 12 baboons in the troop. We have made a note of this in the Methods (page 3). Because this study was conducted at a zoo with socially housed baboons in a large enclosure, recruitment for the study depended on many factors including daily feeding conditions, social events, and motivation.

10. “Page 4, lines 22-23. Some baboons likely exhibit stronger food preferences than others. I’m assuming the same items were used for each baboon, across all trials and both experiments, once their preference was established. Still, it would be helpful to see this explicitly stated. “Depending on the preference of the subject” is ambiguous and could mean that, e.g., Pearl preferred several of the items, and perhaps the items varied for Pearl across tasks.”

We rephrased this (page 4): “We used the same food within each trial, and foods were always either grapes, nuts, or cereal. All animals liked to eat all of these foods. There were no trials where animals did not eat their rewards.”.

11. “Page 6: Relatedly, the data for Pearl are difficult to interpret: Significantly above chance on the sequential condition, unlike Peperella and Sabina. But not significantly above chance in the 1-to-1 subtraction condition experiment, again unlike the other two. IF these tasks are really getting at numerical cognition, why is Pearl so poor at the 1-to-1 subtraction task while excelling at the other tasks, including the ostensibly more difficult sequential task? I realize we can’t know exactly what’s happening with an individual baboon, but Pearl’s performance hints that different strategies may be in effect across the two experiments, and perhaps across baboons.”

Thank you for raising this point. As described above, we have added a random effects regression that measures performance across conditions independently of subject. This regression analysis is more robust than examining individual subject data with binomial tests.

12. “Page 8, line 46. I agree, the range of ratios tested here was quite narrow. Particularly given that the baboons previously exhibited ratio effects for a wider range, I think running the same experiment with different ratios and larger quantities would be helpful.”

Thank you for this suggestion. As we described above, this test is not necessary for making the conclusions we have drawn in the paper and in fact such a test is likely to yield ambiguous results. And, we are unable to conduct trials with a wider range of ratios since losing access to the baboons.

13. “Page 8, line 53. Again, I’m not sure this conclusion is warranted.”

We have addressed this comment in earlier responses. The ad hoc explanations given in this review are insufficient as explanations of the data. We have provided additional data to support our conclusions.

14. “Page 8, line 55, and Page 9, line 4. The claim that “the logic of exact equality develops independently of the concept of exact cardinality from verbal counting” is actually contestable, judging from some of the literature with anumeric adults.”

We rephrased this (page 11): “Thus although the independence of cardinality and counting is debated in the human literature (14,23-30,43), our finding that baboons seem to use a 1-to-1 heuristic to evaluate the relative values of food sets supports the conclusion that they are independent.”

We thank the Reviewers for their helpful feedback because it substantially improved the paper.

Sincerely,

Sarah Koopman

Appendix B

Sarah Koopman

Department of Brain & Cognitive Sciences
University of Rochester
Box 270268, Meliora Hall
Rochester, NY 14627
skoopman@ur.rochester.edu

September 25, 2019

Dear Editor:

Thank you for accepting our revision of manuscript “1-to-1 Correspondence without Language” for publication in *Royal Society Open Science*.

The Reviewers made several additional suggestions based on our revised manuscript. We took this feedback seriously in our second revision of the manuscript. Our revisions are detailed below and **highlighted** in the manuscript.

Reviewer 2

Thank you for your comments. We have added a sentence to the discussion (**page 11**) clarifying that further work is needed to address many of these concerns: “The mechanistic question that remains is how multiple mechanisms for judging quantity – the ANS, parallel individuation, and logical rules like 1-to-1 correspondence – work together during economic and social decision-making to produce coherent behavior.”

We thank the Reviewers for their helpful feedback because it substantially improved the paper.

Sincerely,

Sarah Koopman